# Reconstruction of a generic genome-scale metabolic network for chicken: Investigating network connectivity and finding potential biomarkers

**Ehsan Salehabadi, Ehsan Motamedian[ORCID], Seyed Abbas Shojaosadati***

Biotechnology Group, Department of Chemical Engineering, Tarbiat Modares University, Tehran, Iran

* shoja_sa@modares.ac.ir

**Data Availability Statement:** All relevant data are within the paper and its Supporting Information files.

## Abstract

Chicken is the first sequenced avian that has a crucial role in human life for its meat and egg production. Because of various metabolic disorders, study the metabolism of chicken cell is important. Herein, the first genome-scale metabolic model of a chicken cell named iES1300, consists of 2427 reactions, 2569 metabolites, and 1300 genes, was reconstructed manually based on KEGG, BiGG, CHEBI, UNIPROT, REACTOME, and MetaNetX databases. Interactions of metabolic genes for growth were examined for *E. coli*, *S. cerevisiae*, human, and chicken metabolic models. The results indicated robustness to genetic manipulation for iES1300 similar to the results for human. iES1300 was integrated with transcriptomics data using algorithms and Principal Component Analysis was applied to compare context-specific models of the normal, tumor, lean and fat cell lines. It was found that the normal model has notable metabolic flexibility in the utilization of various metabolic pathways, especially in metabolic pathways of the carbohydrate metabolism, compared to the others. It was also concluded that the fat and tumor models have similar growth metabolisms and the lean chicken model has a more active lipid and carbohydrate metabolism.

## Introduction

Metabolism is an important cellular process in a living cell. Thus, a deep understanding of metabolic networks is required [1]. Collected biological data about metabolic pathways has led us to reconstruct a genome-scale metabolic network that can be mathematically represented [2]. Constraint-based metabolic models are known to be structured models that consider a cell a multi-component system and contain detailed intracellular process information; while because of the black box nature of the models for some cases of machine learning (ML) approaches, further processing may be required to interpret the biological meaning of the model [3]. It should be noted that behaviour or understanding the models derived by ML algorithms is sometimes tough to comprehend or interpret, especially deep neural networks. These ML algorithms are considered examples of black-box models [4]. Therefore, metabolic models

**Funding:** The authors received no specific funding for this work.

**Competing interests:** The authors have declared that no competing interests exist.

will be able to predict the conditions imposed on the cell more reliably [5]. This approach will bring the model prediction as close to reality as possible by considering constraints on the upper bounds of the metabolic reaction fluxes [6]. Besides, metabolic models gain higher predictive power by integration with omics data, which in this respect, their prediction will be more valid than other data-driven models [7]. In recent years, genome-scale metabolic models (GEMs) have been increasingly developed due to the advances in genome sequencing and annotation techniques [8, 9]. GEMs build a bridge between genotypic data and phenotypic traits [10].

The importance of GEMs was strengthened when it was reported that manually and automated metabolic models have been submitted for more than 6200 organisms worldwide. More than 200 of these models belong to eukaryotes [11]. Following the reconstruction of the *saccharomyces cerevisiae* GEM in 2003 as the first eukaryotic model [12], the reconstruction of eukaryotic models became prevalent so that in 2007, the first human GEM RECON1 was reconstructed [2], and human GEMs continued to be updated with the expansion of the network [13], improvement of lipid metabolism [14], energy metabolism [15], and structural information [16]. Other studies in this area include mouse model reconstruction, which is known as the first attempt to reconstruct a mammalian model based on genomic data [17]. The mouse model was also updated in subsequent years using a human model [18]. Chinese Hamster Ovary (CHO) cells are the other interesting eukaryotic models that have been recently reconstructed because of their extensive applications in the biopharmaceutical industries [19, 20].While metabolic networks has been reconstructed for most of the important mammalians in the human life, no attempt has been yet made to reconstruct a metabolic network for chickens as an important source of food. Chickens are important eukaryotes because of their large population in animal husbandries and rural life as well as the annually high consumption of their egg and meat [21]. Studies on chicken first began in 1628 with an investigation on the functions of its arteries and veins. Then, chicken genetic research improved when the chicken's first genetic map was constructed in 1936 [22]. Finally, in 2004, the genome sequencing of chicken wherein scientists had estimated 20000 to 23000 genes for chicken was released [23]. As a potential source of protein, many efforts have been made for both embryonic and adult chickens, ranging from the growth rate to meat yield, and the feeding strategies and efficiencies [24–26]. Studies have shown that breast muscle's *in vivo* glycogen content correlates with meat quality [24]. On the other hand, it has been reviewed that continuous progress in optimizing meat and egg production has led to various disorders in metabolism and reproduction; then it has negative implications on humans as a result of consuming chicken's meat and egg [25]. These obstacles can be overcome by diving more into chicken metabolism. In addition, there are also various metabolic disorders including those with environmental origins, such as oxygen or light regime, feeding strategy, as well as growth-related causes such as extraordinary growth. Some of these common disorders are fatal [27]. Thus, the development of a comprehensive metabolic model can be a platform to study the metabolism of chicken. Such model can also guide us to the treatment and even prevention of various diseases in the chicken.

In this study, for the first time, a comprehensive genome-scale metabolic reconstruction for a chicken cell (named iES1300) was reconstructed. Flux balance analysis (FBA) [28], and single gene deletion as well as double gene deletion analyses were applied to compare robustness of iES1300 and three other important models for growth. Furthermore, transcriptomics data were integrated with iES1300 to construct four types of chicken cell lines, including fat, lean, normal, and tumor. The models were compared to determine essential metabolic differences for growth.

## Material and methods

### The procedure of genome-scale model reconstruction for *gallus gallus*

Fig 1 illustrates the overall procedure of iES1300 reconstruction and building specific models. Using the annotation of genome sequencing for *Gallus gallus* [23], a draft model was generated based on the reconstruction protocol [29]. For this purpose, we applied a pathway-by-pathway analysis of *Gallus gallus* metabolism. The draft consists of all metabolic reactions and their corresponding genes, enzymes, and metabolites collected from the KEGG [30], BiGG [31] and CHEBI [32] databases. To provide the draft of the model, the KEGG database was used to extract biological information of each *gallus gallus* metabolic pathway, including enzymes, reactions, and metabolites [30]. The reaction and metabolite identifiers were selected from the BiGG database to conform to community standards [31]. We also checked metabolites that did not exist in BiGG, their formula, and their charge by using the CHEBI database [32]. If there were not any reactions or metabolite names, we added new names. Regarding the naming of new added reactions and metabolites of metabolic pathways, KEGG reaction identifiers were used. For transport reactions, we used the MetaNetX database identifiers for metabolites and reactions [33]. Mass and charge balance as well as the reversibility of each reaction were also performed. Based on the literature, the intracellular pH of 7.2 was considered for charge balance [34]. This is also a typical pH value in the reconstruction of metabolic networks [29]. Moreover, gene-to-reaction association information was extracted from the related literature and gene orthology obtained from close organisms. Subcellular location information was taken from UNIPROT [35]. Information from CELLO [36] and EukmPLoc v2.0 [37] was also

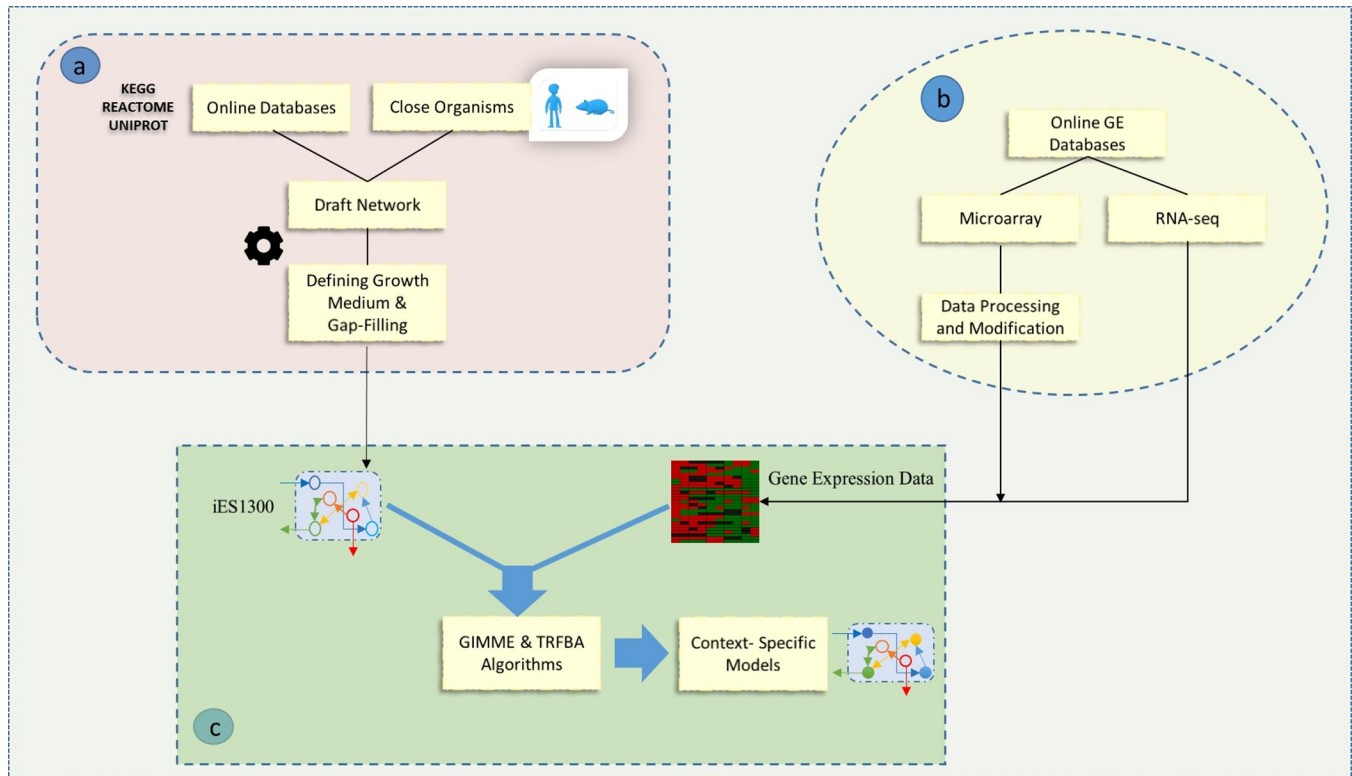

**Fig 1.** Schematic representation of (a) step-by-step genome-scale metabolic reconstruction and (b) using transcriptomics data extracted by online gene expression databases (c) to achieve context-specific models by the integration algorithms.

used for the prediction of cellular location when the localization information was not available in UNIPROT. These two databases use the amino acid sequence of the expressed protein in the FASTA format. CELLO can also use gene nucleotide sequences to predict the location of metabolic reactions. To find transport reactions, information from the REACTOME database was used [38]. Further, regarding lack of data for growth-associated maintenance (GAM) and some of the metabolites related to glycogen and lipid contents that appeared in the biomass formation reaction, these data were selected from the CHO model [19]. Furthermore, the amount of non-growth associated maintenance (NGAM) was taken from a mouse cell [39]. Also, for the biomass reaction generation, the existing information for chicken, including amino acid percentage and DNA components percentage was used. In addition, amino acid coefficients information was taken from the High-performance Integrated Virtual Environment (HIVE) database [40] and nucleotide information of the chicken genome data in NCBI [41]. More details about biomass reaction are available in the S2 File. The gap-filling process was also established so that the added reactions made the model capable of growing.

### *In silico* simulations condition

To solve linear programming problems, Constraint-Based Reconstruction and Analysis (COBRA) toolbox in MATLAB 2017b software and the glpk solver package were utilized [42]. COBRA toolbox is a MATLAB package and has been developed to implement constraint based reconstruction and analysis methods; i.e. it allows researchers to infer and analyze reconstructed models [42]. In this way, COBRA toolbox has the capability of converting biological data into a mathematical model, then evaluate models by defining or changing objective functions, and improve the reconstructed model by deploying gap-filling process [42]. For intracellular reversible reactions, lower and upper bounds were set at -1000 and 1000 $\frac{mmol}{gDCW.h}$, respectively. Contrarily, for intracellular irreversible reactions, lower and upper bounds were set at 0 and 1000 $\frac{mmol}{gDCW.h}$, respectively. The upper bound of all the exchange reactions was set at 1000 $\frac{mmol}{gDCW.h}$. It is conventional that all reactions in metabolic models must have infinite lower and upper bounds to accept any flux. This infinity is shown by setting 1000 $\frac{mmol}{gDCW.h}$, as 1000 is considered to be a huge and infinite flux amount. An RPMI-like culture medium was also selected for the simulation of the medium, as it is proved to be one of the reliable culture media for growing chicken cells with respect to the studies on various types of chicken cells [43–45], and the lower bounds of exchange reactions were fixed based on [46]. Detailed components of the RPMI-like culture medium are represented in the S2 File. Besides, the biomass reaction was selected as the objective function in all of the simulations.

### Comparison of the network

To date, a large number of studies have used different methods of network topology analysis to evaluate metabolic networks in terms of phylogenetic relationships [47]. Therefore, topology analysis of iES1300 was applied to see how phylogenetically close it is to its peers. We performed the single- and double-gene deletion analyses to compare iES1300 with the metabolic models of human (RECON1) [2], *Saccharomyces cerevisiae* (iMM904) [48], and *Escherichia coli* (iJO1366) [49] to evaluate all metabolic networks in terms of phylogenetic relationships [47]. It has to be mentioned that Single Gene Deletion is one of the COBRA toolbox analysis modules and calculates the effect of each gene on the cell growth one-by-one [42]. For Double Gene Deletion, the same procedure is also performed to remove the double genes. These analyses indicated that iES1300 was more flexible and robust compared to other prokaryotic and eukaryotic models. GR ratio (predicted growth rate after gene deletion per growth rate for

wild type) was also applied to indicate the efficacy of single- or double-gene deletion. According to the method presented by [50], we defined a threshold on these growth rate ratios. In such a way that this ratio was lower than $10^{-5}$ for a particular gene, it must be a lethal gene, and if this ratio was just below 1 for other specific genes, it meant that those genes are categorized as sick genes. Therefore, we determined sick and lethal genes based on the single-gene deletion analysis, while the interactions of genes, synthetic lethal, and synthetic sick genes were specified based on the double-gene deletion analysis. Finally, to compare the four models, the results of the single-gene deletions and the number of interactions were normalized by dividing the number of genes in each model, whereas the results of the double-gene deletions were divided by the square of the number of genes in each model.

## Integration of gene expression data

Considering HCC as an important disorder in the chickens liver [51], and the negative effects of adiposity on the economics of the poultry industries, especially in the case of meat quality [52], gene expression data from four species of chicken, including chicken liver control samples of hepatocellular carcinoma [53], normal cell lines [54], and adipose tissue samples of lean and fat [55] were taken from ArrayExpress, which is one of the largest online databases that acts as an archive of functional genomics data [56]. It was founded for microarray datasets at first, but the number of datasets submitted based on sequencing experiments has surprisingly been exceeded [56]. In our previous research [57], we indicated that TRFBA [58] and GIMME [59] are successful algorithms, especially for prediction of growth. So, both algorithms were applied to generate high-quality context-specific models. First, GIMME was used to remove reactions supported by genes with low expression levels. Then, TRFBA was employed to constrain the upper bound of the remaining reactions in the model according to the expression level of their supporting genes. In fact, TRFBA first converted all of the reversible reactions of a metabolic model into irreversible and "withoutOR". Next, it added a set of constraints to limit the rate of reactions [58] as follows:

$$\sum_{i \in K_j} v_i \leq E_j \times C \qquad (1)$$

Where $v_i$ is the reaction flux of i, $E_j$ is the expression of the gene j, $K_j$ is the set of indices of reactions supported by metabolic gene j, and C is a constant parameter that converts the expression levels to the upper bounds of the model reactions. This coefficient indicates the maximum rate supported by one unit of expression level of a gene; thus, the unit for C is mmol gDCW$^{-1}$ h$^{-1}$.

The threshold value was set to the 25th percentile of the given expression data to inactivate reactions below a specific mRNA transcript level based on Machado and Herrgard study [60]. The second parameter for the objective function flux cutoff was used according to the original paper, i.e., the GIMME output context-specific metabolic network was forced to grow no less than 90% of the maximum growth [59]. The parameter of TRFBA (C) for each cell line was also changed in a stepwise approach according to the method presented in the next section.

## Differentiation of cell lines using principal component analysis

After applying GIMME, the stepwise TRFBA was employed by stepwise change in C similar to the method presented by [57]. C was changed from zero to $C_{brk}$ with a step size of 0.1 of $C_{brk}$, hence, nine flux distributions were constructed for each cell line. $C_{brk}$ is the point at which the growth rate does not change with an increase in the value of C [57]. In fact, TRFBA was used to maximize the growth rate for each cell. On the other hand, to avoid the well-known degeneracy of solutions, the Manhattan norm of the flux distribution was minimized while the

optimal growth rate was given as constraint [61]. Correlated reactions with growth were determined for each cell line by calculating the Pearson correlation coefficient between each reaction flux and growth rate so that reactions with a coefficient more than 0.9 (P-value $\leq$ 0.05) were considered correlated. In the next step, the common growth-correlated reactions for the four cell lines were selected, and PCA was performed to differentiate the cell lines using the selected reactions.

## Results and discussions

### Characteristics of the reconstructed model

The reconstructed model contained 2427 biochemical reactions from 95 metabolic subsystems, 1300 genes, and 2569 metabolites. Of these reactions, 1910 reactions were gene-associated reactions and 295 of them were non-gene-associated. None-gene associated reactions, also known as orphan reactions, are those that have not yet been identified which genes or proteins are encoding them [62], but we added them to the model so that the model can grow. On the contrary, gene associated reactions are reactions for which a gene or several genes have been assigned [62]. Even the most comprehensive reconstructed models with highest confidence scores may have their own deficiencies [62]. This is mainly due to inadequate knowledge of the metabolism which leads to missing metabolites and reactions. For this reason, gap-filling is needed to refine the model inconsistencies [62]. To this end, for the growth of the chicken cell, we checked each component in the biomass production reaction to make sure they are not dead-end metabolites and see if they carry flux. During the gap-filling process, 67 reactions were added to the model to make it capable of growing. Fig 2A categorizes the reactions of iES1300 into nine main subsystems. As shown in this figure, among these subsystems, lipid and energy metabolism have the largest and the smallest distributions, respectively. Fig 2B, on the other side, determines that iES1300 consists of 10 subcellular locations, named cytosol, mitochondrion, extracellular space, endoplasmic reticulum, Golgi apparatus, lysosome, peroxisome, cytosolic membrane, endosome, and nucleus. This figure also shows that cytosol has the largest metabolite distribution. In Fig 2C, we can see similar as well as different reactions of iES1300 that compared with the two important mammalian metabolic GEMs. Fig 2C indicates that iES1300 has a relatively high similarity with the Recon2v4 and iCHOv1 networks. The new reactions of iES1300 may refer to the HMR reactions existing in the Recon3D reaction list used in iES1300, spontaneous reactions and also the reactions with different subcellular locations. Moreover, comparison of gene-associated reactions of iES1300 with three GEMs (Fig 2D) indicated that iES1300 had a lower ratio of non-gene-associated reactions to the total number of reactions of each model. To this end, we counted the number of reactions that had gene to reaction association and divided it by the number of all reactions in each model. Pseudo reactions are also referred to the exchange, demand, sink, biomass, and ATP maintenance reactions.

### Comparison of iES1300 with the other eukaryotic and prokaryotic models

Three different types of organism cells were chosen to be compared with iES1300 using the single- and double-gene deletion analyses. Table 1 presents that the effect of single- and double-gene deletions on growth of multicellular organisms are much lower than those of unicellular organisms. The lower GR ratio of genes with interactions in two multicellular species also showed that compared with unicellular models, multicellular ones had more robustness and flexibility to genetic perturbations. In addition, the analyses confirmed that the trait of iES1300 was similar to the eukaryotic models.

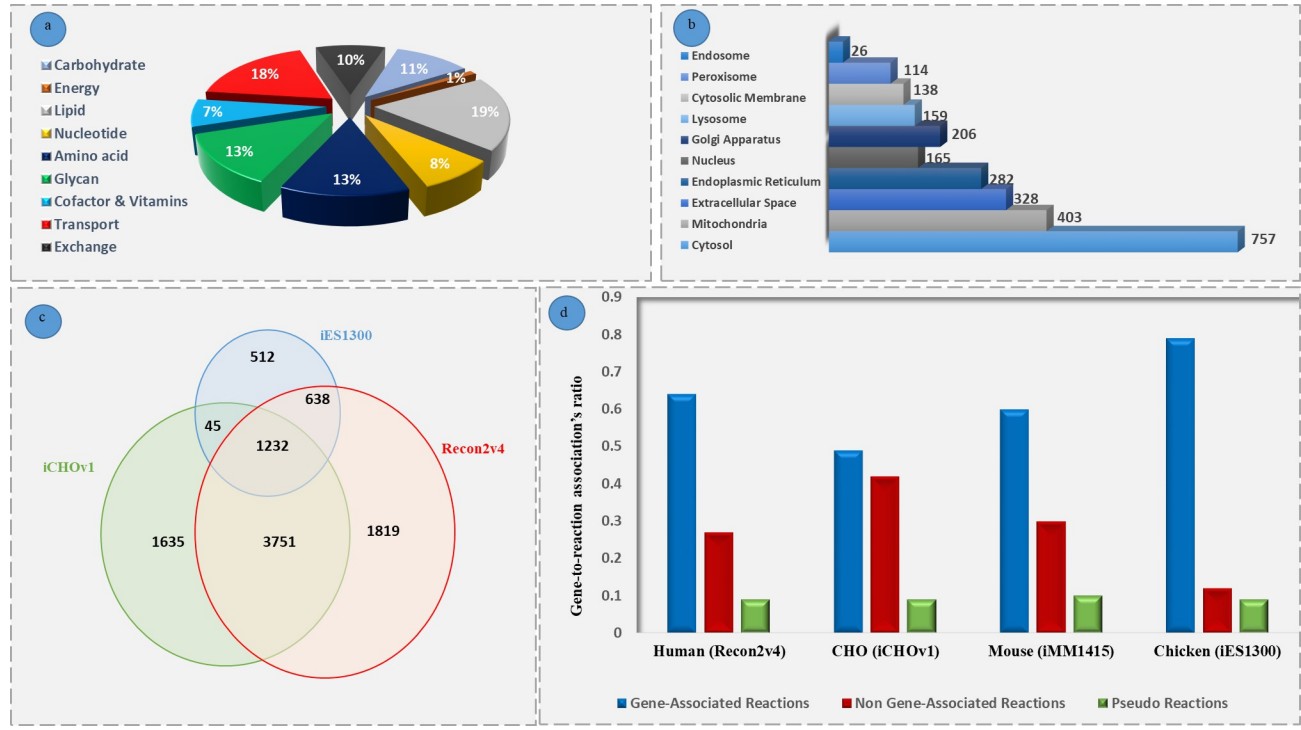

**Fig 2. Characteristics of the reconstructed model.** (a) In iES1300, there are 9 main metabolism categories of which the lipid metabolism is responsible for the largest reaction distribution. (b) Cytosol is considered to have the largest metabolite distribution. (c) Investigating the number of reactions shared by three important mammalian models. The newly added reactions in iES1300 in comparison to the two other models can be discovered. (d) Examination of the gene-to-reaction association's ratios (To the total number of reactions of each model) in the four important mammalian models exhibits that iES1300 has much lower orphan reactions compared to the other models.

## Evaluation of metabolic similarities for cell lines

Growth vs. $C/C_{brk}$ for each model is presented in Fig 3. In this figure, we can see that the growth patterns of fat and tumor cell models are nearly analogous. In addition, metabolisms of lean model is more susceptible to the change of C compared to normal.

Furthermore, by applying PCA for the common growth-related reactions, it was found that the normal and lean models were significantly different from tumor and fat models (Fig 4A). The PCA results also indicated that the normal chicken cell was the most different cell line from the lean, fat, and tumor cell lines. Fig 4B illustrates that by using the first principal component, 85% of the difference between normal cell metabolism and other cell lines can be explained. We further presented the difference of lean chicken cell with other cell lines by the second principal component with approximately 15% variance. It can be seen in Fig 4A that the fat model has metabolic similarities to tumor model. Therefore, we can state that the

**Table 1. Results of single and double gene deletion analysis on four metabolic models using FBA approach.**

| Cell lines | Number of Genes | Ratios | | |
|---|---|---|---|---|
| | | Growth-Related Genes | Growth-Related Double-Genes | Genes with Interaction |
| iES1300 | 1300 | 0.05 | 0.0007 | 0.09 |
| RECON1 | 1905 | 0.056 | 0.0002 | 0.07 |
| iMM904 | 905 | 0.16 | 0.0014 | 0.016 |
| iJO1366 | 1367 | 0.21 | 0.0013 | 0.15 |

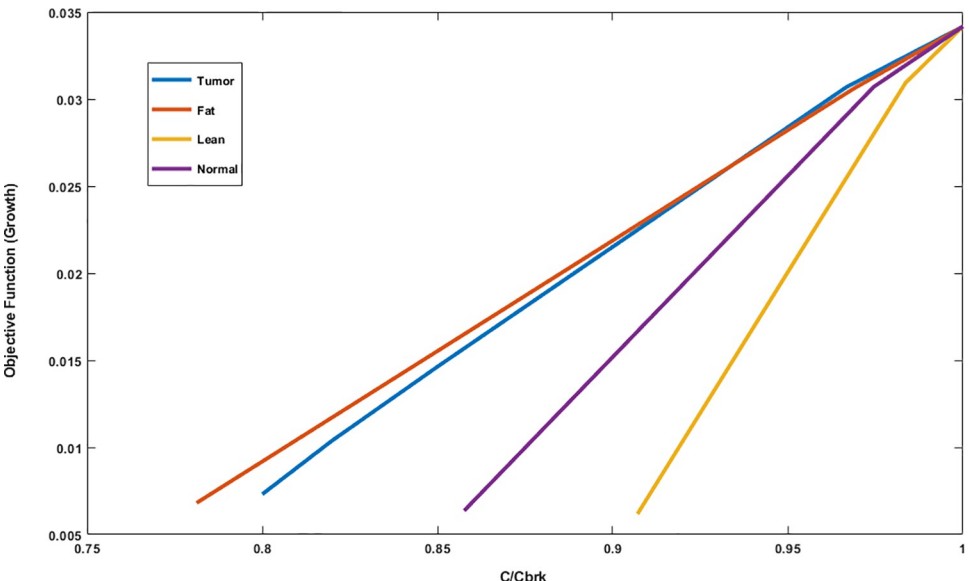

**Fig 3. The different patterns of the growth sensitivity to the normalized parameter of TRFBA algorithm (C/C$_{brk}$), are a source for differentiation of the four models.**

normal model had key metabolic differences from lean, fat, and tumor chick cell metabolisms. Focusing on these differences and targeting them can prevent such metabolic disorders in chickens. Results of PCA also verified that a considerable number of reactions that shifted along the PC2 were orthogonal to the reactions that appear in PC1, indicating that the lean and the normal models have presumably separate growth mechanisms.

Fig 4C also shows the 28 main reactions in the differentiation of fat, lean, tumor, and normal chicken cell lines. Reactions with low values of absolute principal components were not considered because their differentiated flux distributions were insignificant.

Besides, it is worth mentioning that the fat and tumor chicken models had strong resemblances, mostly because of their glucose, nucleotide, and lipid metabolism activities and many essential amino acid exchange reactions of these two models. These results are presented in the S2 File, where all the differentiated reactions in PCA are explained. Research works, especially in the field of obesity and Hepatocellular Carcinoma (HCC), indicated that in obesity, fat accumulation leads to liver malfunction, and consequently the liver cannot send out more triglycerides by very-low-density lipoprotein (VLDL) than that are synthesized. This intrahepatic triglyceride increasing would, in turn, result in fatty liver, and consequently liver failure and HCC [63, 64]. In the case of iES1300, lots of relationships and similarities were observed, especially in three pathways of lipid metabolism, including Fatty Acid Synthesis (FAS), sphingolipid, and glycerophospholipid metabolisms. Similarities in the flux patterns of FAS could lead both tumor and fat models to equally produce Palmitoyl-COA, which is a key metabolite in the progression of many other lipid pathways. Sphingolipid metabolism is one of the metabolic pathways, which is affected by FAS. This metabolic pathway is known for having some bioactive metabolites involved in the regulation of cell growth [65]. Therefore, it can be one of the primary sources of similarity in both models. Tracking sphingolipid and glycerophospholipid metabolisms has also revealed that some of their major metabolites play a key role in the biomass objective function reaction. Given that the nucleotides are widely used in various functions of all cells, and because of their relation with cell proliferation to DNA replication and RNA production [66], the more balance in the nucleotide metabolism activity could result in

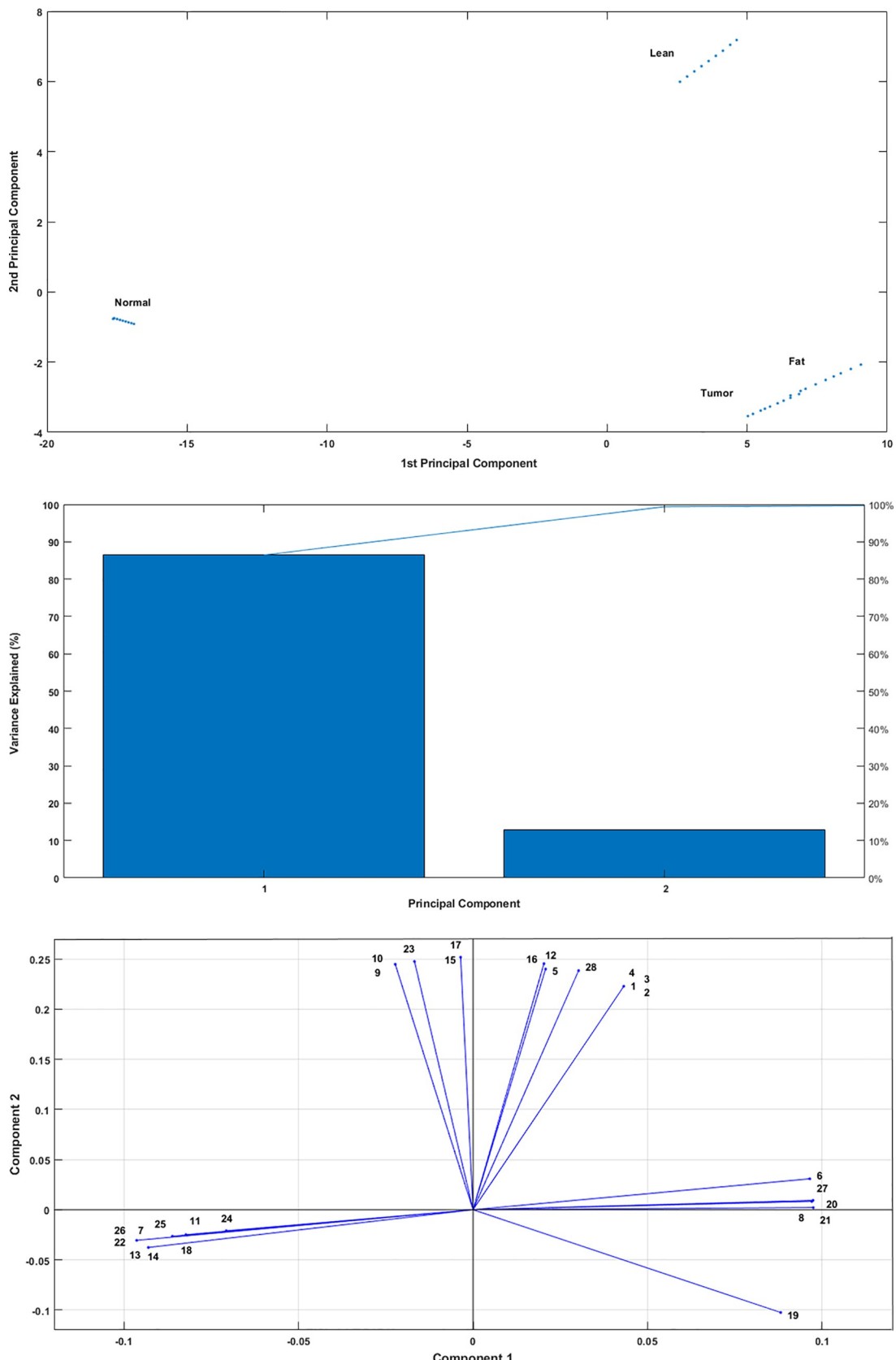

**Fig 4. PCA for extracting the reactions that play the role of biomarkers.** (a) PCA shows that the 36 flux distributions created by the combination of two integration algorithms, GIMME[59], and stepwise TRFBA[58] are successfully categorized into four different groups so that each group belongs to one cell line. (b) The variance explanation chart indicates that the 1st principal component is responsible for about 85% of the variance. The other 15% can be explained by the 2nd principal component. (c) The main differentiated reactions in PCA. 28 reactions are shown to have participated in the differentiation of the fat, lean, normal and tumor cells. Other reactions with lower absolute PC values were not investigated. The complete list of the reaction names is available in the S2 File.

more equivalency of nucleotide production used for biomass generation. Moreover, it has been proved that obesity can systemically impact glucose metabolism by elevating glucose and insulin level, which favors cancer cell progression [67]. It can be presumed that the similarity in these three main metabolisms (glucose, nucleotide, and lipid metabolisms) controlling cell growth and proliferation affects the similarity between the amounts of many biomass reactants produced in tumor and fat models. This similarity could lead to the identical cell growth patterns in the tumor and fat models rather than the normal and lean models.

PCA revealed that in the 1st Principal Component, which differentiated the normal chicken model from other models, especially tumor model, there were numerous distinguished reactions in three metabolisms, including nucleotide, carbohydrate, and lipid metabolism. Although number of these reactions were higher in the tumor cell metabolism than in the normal cells, the results demonstrated that glycolysis and pentose phosphate pathways were the most critical metabolic pathways in diversities of normal and tumor chicken models. It is also noteworthy that PC1 results of the 28 leading differentiated reactions indicated that the glycolysis pathway in tumor cell metabolism had higher activity toward the production of 3-phosphoglycerate, which is a key metabolite in the development of glycine, serine, and threonine metabolism. On this account, the tumor cells are expected to be more active than a normal cell in this metabolic pathway. This assertion could be justified by knowing that serine and glycine provide the main precursors for tumor cell metabolism [68]. It has been also observed that despite the fact that most reactions happen in the tumor cell, the normal cell could produce significant amounts of ribose-5-phosphate. Further investigations have shown that in the next step, this metabolite is converted to 5-phospho-ribose 1-diphosphate, which plays an influential role in the progression of nucleotide metabolism. It is important to note that nucleotides can be synthesizable from two main metabolisms of *de novo* synthesis and salvage pathways [66]. The proliferating cells such as cancer cells are more eager to synthesize their required nucleotides through *de novo* synthesis [69]. Similarly, in the present study, iES1300 demonstrated that the preference of normal chicken cell to use recycling of its nucleosides and nucleobases through salvage pathways is much higher than tumor cell.

On the other side, for the 2nd Principal Component, which differentiated the lean cell line model from the others, especially fat model metabolism, various differentially flux distributions were observed. Results illustrated that in many metabolisms, because of several positive shifts along the PC2 axis, the metabolic activity of lean chicken can be more than that of fat chicken. However, a few of them were identified to be significant. Intriguingly, we perceived that the activity of enzyme phosphoglucomutase in the production of glucose-6-phosphate and the way it is used can be the dominant source of differences between lean and fat models. In the lean model, a considerable amount of glucose-6-phosphate heads towards inositol phosphate as well as ascorbate and aldarate metabolisms to produce significant amounts of uridine diphosphate glucose (UDPG). Since UDPG is a key precursor in starch and sucrose metabolism, it was expected that the lean model had much more activity on this pathway. Additional investigations indicated that the increase in activity of lean model was not due to glycogen production and storage, but it resulted from the production of glucose by 4-alpha-

glucanotransferase. As a result, the starch and sucrose metabolism was considered a key metabolic pathway, and the principal source of cell metabolism for glucose generation and utilization. Glucose is consumed in the lean model at a significantly higher rate, even in adipose tissue. Previous studies on the relationship between glycogen and lipid oxidation in the liver and muscles have also shown that the glycogen storage reduction could increase lipid oxidation by stimulation of cellular energy state [70, 71]. Likewise, in obese adipocyte cell metabolism, the lower gene expression of fatty acid pathways because of defection in mitochondrial function resulting from a decrease in the mitochondrial acetyl-CoA concentration has already been established [14]. In this research work, iES1300 represented a significant diversity in the flux distributions of the lean and fat models, especially flux of three important reactions in cholesterol metabolism, fatty acid oxidation, and glyoxylate and dicarboxylate metabolism are significantly different. These reactions contributed to the production of mitochondrial acetyl-CoA and subsequently significant discrepancy in fatty acid oxidation metabolism was observed.

## Conclusion

Chickens are the animals most associated with humans in rural life and animal husbandries to produce meat and egg. Therefore, understanding the chicken cell's metabolism is beneficial to raise healthier chickens with a better feeding strategy and a higher meat quality. By reconstructing a consensus metabolic model for a chicken cell, one can deeply comprehend a chicken's metabolism. The reconstructed model can then be used as a platform for other researchers to broaden their knowledge and studies of chicken cells' metabolic interactions more efficiently. Accordingly, to reconstruct the first genome-scale metabolic model of the chicken cell, chicken's biological and genomic data were manually collected in the form of a draft from different available bioinformatics databases. Afterwards, the model was curated by performing the gap-filling process. The final model consisted of 2427 reactions, 2569 metabolites, and 1300 genes. The chicken model was compared with three other important models to evaluate the interaction of metabolic gene networks, and this comparison demonstrated the relative similarity of the chicken's gene network to human. After model reconstruction, the transcriptomics data of the four cell types of lean, fat, normal, and tumor were integrated using the two algorithms of GIMME and TRFBA. Finally, implementing PCA, we concluded that PCA has appropriately differentiated cell types from each other and recommended essential biomarkers. These biomarkers primarily participate in different metabolic pathways, such as carbohydrate metabolism, to distinguish normal cells from three other cell lines. Therefore, in addition to a general platform, we highlighted potential biomarkers that drugs can target to avoid chickens' common metabolic diseases.

## Supporting information

**S1 File. The Excel and SBML version of the reconstructed model.** In this supplementary file, a zip file containing the Excel and SBML version of the reconstructed chicken cell model is attached.
(RAR)

**S2 File. Detailed information of the model.** In this supplementary file, detailed information in the form of Excel sheets that has been mentioned in the paper including protein and DNA components coefficients, simulated growth medium, all and the main differentiated reactions from PCA results are available.
(XLSX)

## Acknowledgments

The authors thank Tarbiat Modares University Research and Technology Unit for supporting this study.

## Author Contributions

**Conceptualization:** Ehsan Salehabadi, Ehsan Motamedian, Seyed Abbas Shojaosadati.

**Data curation:** Ehsan Salehabadi, Ehsan Motamedian.

**Formal analysis:** Ehsan Salehabadi, Ehsan Motamedian.

**Investigation:** Ehsan Salehabadi, Ehsan Motamedian, Seyed Abbas Shojaosadati.

**Methodology:** Ehsan Motamedian, Seyed Abbas Shojaosadati.

**Project administration:** Seyed Abbas Shojaosadati.

**Resources:** Seyed Abbas Shojaosadati.

**Software:** Seyed Abbas Shojaosadati.

**Supervision:** Ehsan Motamedian, Seyed Abbas Shojaosadati.

**Validation:** Ehsan Salehabadi, Ehsan Motamedian.

**Writing – original draft:** Ehsan Salehabadi.

**Writing – review & editing:** Ehsan Motamedian, Seyed Abbas Shojaosadati.

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
