## [Decision Letter · Decision Letter 0]

28 Dec 2021

PONE-D-21-17962Reconstruction of a generic genome-scale metabolic network for chicken: investigating network connectivity and finding potential biomarkersPLOS ONE

Dear Dr. Shojaosadati,

Thank you for submitting your manuscript to PLOS ONE. After careful consideration, we feel that it has merit but does not fully meet PLOS ONE’s publication criteria as it currently stands. Therefore, we invite you to submit a revised version of the manuscript that addresses the points raised during the review process.

We look forward to receiving your revised manuscript.

Kind regards,

Mingzhi Liao

Academic Editor

PLOS ONE

Journal Requirements:

3. Please upload a new copy of Figures 1, 2, 3 and 4 as the detail is not clear. Please follow the link for more information: " ext-link-type="uri" xlink:type="simple">https://blogs.plos.org/plos/2019/06/looking-good-tips-for-creating-your-plos-figures-graphics/"
" ext-link-type="uri" xlink:type="simple">https://blogs.plos.org/plos/2019/06/looking-good-tips-for-creating-your-plos-figures-graphics/"

Reviewers' comments:

Reviewer's Responses to Questions

**Comments to the Author**

1. Is the manuscript technically sound, and do the data support the conclusions?

Reviewer #1: Yes

Reviewer #2: Yes

2. Has the statistical analysis been performed appropriately and rigorously? 

Reviewer #1: Yes

Reviewer #2: Yes

3. Have the authors made all data underlying the findings in their manuscript fully available?

Reviewer #1: Yes

Reviewer #2: Yes

4. Is the manuscript presented in an intelligible fashion and written in standard English?

Reviewer #1: Yes

Reviewer #2: Yes

5. Review Comments to the Author

Reviewer #1: Your interest of research is good. Revise your manuscript accordingly and revise the grammatical errors in the paper explain all the possible terms mentioned in the paper as they are described not properly through the manuscript

Reviewer #2: In this manuscript, the authors reconstructed a generic genome-scale metabolic network for chicken named iES1300, which was consists of 2427 reactions, 2526 metabolites, and 1300 genes. In addition, they compared the iES1300 with three other important models to evaluate the interaction of metabolic gene networks. Meanwhile, they found several important biomarkers based on PCA. Overall, this work is meaningful and well-organized.

However, the whole manuscript has bioinformatics results only. In order to illustrate the reliability of those biomarkers, in vitro or in vivo biochemical verification is suggested.

1.In“2.1 The procedure of genome-scale model reconstruction for gallus gallus” part, it will be helpful to explain the rules and the basis for adding new name for those reactions or metabolite which have no name.

2.In “2.2 In silico simulations condition”, is there any reference for the setting of the reaction upper and lower bounds?

3.In “3.1 Characteristics of the reconstructed model” part, it is better to add the calculation of network correlation to illustrate the similarity between the networks.

4.In the discussion part, it's suggested to include some critical thinking and exemplify the application of this research.

5.Please pay attention to the inconsistency in font and case, such as “by” in line 234, and “Metabolisms” in line 267.

6. PLOS authors have the option to publish the peer review history of their article (what does this mean?). If published, this will include your full peer review and any attached files.

Reviewer #1: No

Reviewer #2: No

---

## [Author Response · Author response to Decision Letter 0]

19 Feb 2022

We would like to greatly appreciate your attention to our manuscript. We acknowledge that our manuscript is reinforced after the revisions applied to the manuscript based on the constructive comments received from the professional editor and reviewer. As requested, the questions and comments are answered and addressed below meticulously. We also used both highlighting and track changes. The edited manuscript is attached, as well. We hope the major and minor revisions serve to draw your attention to find the manuscript worthy of publication. 

Thank you again for perusing this context. We would welcome any further specific questions you might have.

---

## [Editor Report · Decision Letter 1]

8 Mar 2022

Reconstruction of a generic genome-scale metabolic network for chicken: investigating network connectivity and finding potential biomarkers

PONE-D-21-17962R1

Dear Dr. Shojaosadati,

We’re pleased to inform you that your manuscript has been judged scientifically suitable for publication and will be formally accepted for publication once it meets all outstanding technical requirements.

Kind regards,

Mingzhi Liao

Academic Editor

PLOS ONE

Additional Editor Comments (optional):

All issues have been solved.
---

## [Editor Report · Acceptance letter]

12 Mar 2022

PONE-D-21-17962R1 

Reconstruction of a generic genome-scale metabolic network for chicken: investigating network connectivity and finding potential biomarkers 

Dear Dr. Shojaosadati:

I'm pleased to inform you that your manuscript has been deemed suitable for publication in PLOS ONE. Congratulations! Your manuscript is now with our production department. 

Kind regards, 

on behalf of

Dr. Mingzhi Liao 

Academic Editor

PLOS ONE